# Genetic Characterization of a New HIV-1 Sub-Subtype A in Cabo Verde, Denominated A8

**DOI:** 10.3390/v13061093

**Published:** 2021-06-08

**Authors:** Rayana Katylin Mendes Da Silva, Isabel Inês Monteiro de Pina Araujo, Karine Venegas Maciera, Mariza Gonçalves Morgado, Monick Lindenmeyer Guimarães

**Affiliations:** 1Laboratório de AIDS e Imunologia Molecular, Instituto Oswaldo Cruz, Fiocruz, Rio de Janeiro 21040-360, Brazil; rayanakmendes@gmail.com (R.K.M.D.S.); karine_948@hotmail.com (K.V.M.); mmorgado@ioc.fiocruz.br (M.G.M.); 2Faculdade de Ciências e Tecnologia, Universidade de Cabo Verde, Praia 379C, Cape Verde; iniza.araujo@adm.unicv.edu.cv

**Keywords:** HIV-1 diversity, sub-subtype A, surveillance, Cabo Verde

## Abstract

Previous molecular characterization of Human immunodeficiency virus (HIV-1) samples from Cabo Verde pointed out a vast HIV-1 *pol* diversity, with several subtypes and recombinant forms, being 5.2% classified as AU-*pol*. Thus, the aim of the present study was to improve the characterization of these AU sequences. The genomic DNA of seven HIV-1 AU *pol*-infected individuals were submitted to four overlapping nested-PCR fragments aiming to compose the full-length HIV-1 genome. The final classification was based on phylogenetic trees that were generated using the maximum likelihood and bootscan analysis. The genetic distances were calculated using Mega 7.0 software. Complete genome amplification was possible for two samples, and partial genomes were obtained for the other five. These two samples grouped together with a high support value, in a separate branch from the other sub-subtypes A and CRF26_A5U. No recombination was verified at bootscan, leading to the classification of a new sub-subtype A. The intragroup genetic distance from the new sub-subtype A at a complete genome was 5.2%, and the intergroup genetic varied from 8.1% to 19.0% in the analyzed fragments. Our study describes a new HIV-1 sub-subtype A and highlights the importance of continued molecular surveillance studies, mainly in countries with high HIV molecular diversity.

## 1. Introduction

Human immunodeficiency virus (HIV) originated from multiple zoonotic transmission events of the simian immunodeficiency virus (SIV) from non-human primates to humans in Central and West Africa, resulting in two types of HIV-1 that encompassed groups M, N, O, and P [1,2,3], in addition to HIV-2 (groups A–H) [4]. The HIV-1 M group is responsible for the HIV/AIDS pandemic and, currently, phylogenetic analyses based on complete genomes revealed that this group is composed of 10 subtypes (A, B, C, D, F, G, H, J, K, L), as well as circulating (CRFs) or unique (URFs) recombinant forms [5,6]. In 2018, a reclassification of HIV-1 A sub-subtypes was proposed based on phylogenetic analyses carried out with full-genome sequences obtained from public sequence databases. Thus, subtype A was subdivided into six sub-subtypes (A1–A4, A6, A7) [7]. Sub-subtype A5 was not found in its pure form, but is still part of the CRF26_A5U [8].

The HIV pandemic remains a global public health problem, with 38 million people living with HIV (PLWH) in 2019 [9]. Located in West Africa, Cabo Verde has about 2500 (2100–3000) PLWH, which is equivalent to 0.6% of the population between 15 and 49 years old [9]. In a previous study of HIV samples from Cabo Verde, molecular epidemiological data of the *pol* region revealed a high prevalence of HIV-1 subtypes G (36.6%) and CRF02_AG (30.6%), URFs (10.4%), sub-subtype F1 (9.7%), B (5.2%), CRF05_DF (3.0%), C (2.2%), CRF06_cpx (0.7%), CRF25_cpx (0.7%), and CRF49_cpx (0.7%), while all HIV-2 infections belonged to group A. Moreover, a complex profile of drug resistance mutations occurring in almost 48% of the CV HIV-1 positive individuals under cART, considering any class of antiretroviral drugs [10]. Among the samples classified as URFs, 5.2% had the same AU pol profile and were grouped in a monophyletic cluster with high support [10]. Thus, the aim of the present work was to obtain and characterize the HIV-1 full-length genome sequences, which allowed the description of a new HIV-1 sub-subtype A, here denominated as A8.

## 2. Materials and Methods

### 2.1. Study Population 

Previous studies from our group recruited a total of 169 individuals living with HIV-1 in Cabo Verde from 2010 to 2011. HIV-1 pol sequences (covering the protease (PR) and partial reverse-transcriptase (RT), positions 2253–3251 relative to the HXB2 genome) were available for 134 of them. A highly significant supported AU-pol cluster including seven sequences corresponded to 5.2% of the classified sequences [10], which were the focus of the present study.

### 2.2. Amplification of HIV-1 Full-Length Genomes and Phylogenetic Analyses

Only whole blood aliquots were available and stored since sample collection in 2010/2011 at −20 °C and were used in these analyses. Genomic DNA was extracted from individuals’ whole blood using a QIAamp DNA Blood Mini Kit (QIAGEN, Qiagen, Hilden, Germany). The double-stranded proviral DNA was amplified using nested-PCR employing Platinum Taq DNA polymerase (Invitrogen, Carlsbad, CA, USA) into four overlapping fragments (i.e., fragment 1 (408–2594), fragment 2 (2253–4830), fragment 3 (4653–7811), and fragment 4 (6954–9625) relative to the HXB2 genome), using HIV-1 specific primers as described in Reis et al. (2017) [11]. Nucleotide sequences obtained in the present work are available from the GenBank database under accessions (MW353966–MW353970).

The amplified products were purified using the Illustra GFX PCR DNA and gel Purification Kit (GE Healthcare, Little Chalfont, Buckinghamshire, United Kingdom) and sequenced on an ABI 3130 Genetic Analyzer using the ABI BigDye Terminator v.3.1 Cycle Sequencing Ready Reaction kit (Applied Biosystems, Foster City, CA, USA). The chromatograms were analyzed and edited using the Seqman software from the package DNASTAR Lasergene (DNAStar, Madison, WI, USA). 

The phylogenetic trees of maximum likelihood (ML) were reconstructed with PhyML version 3.0 [12] using the general time reversible (GTR) model of nucleotide substitutions. The approximate likelihood ratio test (aLRT) was used to estimate the confidence of the branch in the tree. The phylogenetic trees reconstructed were visualized and edited using the Figtree software version 1.4.4 [13]. Reference sequences of HIV-1 group M subtypes (A–D, F–H, J–L), sub-subtypes (A1–A4, A6, A7-, F1–F2), and CRF26_A5U sequences were obtained from the Los Alamos HIV database [5]. The sequence data sets were obtained by grouping our sequences and the reference sequences. 

A basic local alignment search tool (BLAST) [14] was performed in order to identify sequences with high similarity to the studied sequences. We investigated the complete genome and pol region. The retrieved sequences were included in phylogenetic analyses.

Recombination analyses were performed using a bootscan implemented in Simplot v3.5.1 software with the following parameters: 400 nt window, 20 nt increments, and NJ method under Kimura’s two-parameter correction with 100 bootstrap replicates [15].

### 2.3. Genetic Distances

The genetic distances among the analyzed sequences (intragroup distance) and between these and the other A sub-subtypes (intergroup distance) were calculated using the Mega version 7.0 [16] software for each analyzed fragment (complete genome 803–9496, gag (803–2280), PR/RT (2065–5100), and env (6223–8804) relative to the HXB2 genome).

### 2.4. Drug Resistance Analysis (DRM)

Analysis of PR/RT resistance mutations was performed through the Stanford HIV Drug Resistance Database website. HIV Database for Transmitted DRM-TDRM (CPR Tool version 9.0) and DRM (HIVdb Program version 6.3.1) for naive and treated patients, respectively [17,18].

## 3. Results

### 3.1. Sociodemographic and Clinical Data

Among the seven HIV-1 AU samples investigated in Cabo Verdean individuals, six were from Santiago and one was from the Sal Island. Five were females and two were males. Only two patients had a known epidemiological linkage resulting from mother-to-child transmission (Table 1). Considering the drug resistance mutation profiles, just CV.10.105 and CV.10.115 presented NNRTI DRM, whereas all of them presented L10I or L10V minor PI DRM.

### 3.2. Genome Amplification and Sequence Analysis

From these previously classified HIV-1 AU pol samples, it was possible to amplify and sequence the complete genome for the two of them (CV.10.115-864 to 9615, CV.10.126- 413 to 9516 relative to the HXB2 genome), which were obtained from patients without epidemiological linkage. Due to the low amount of the biological material available, only partial genome sequences could be obtained for three individuals (CV.10. 105- 2254 to 6533, CV.11.270- 985 to 5565, and CV.11.290- 1339 to 5767 relative to the HXB2 genome) and the two remaining ones. Only the initial fragment of PR/RT obtained at the original study was investigated (CV.10.164 and CV.11.275–2253 to 3218 relative to the HXB2 genome) (Table 1). 

The ML tree from the complete genomes showed that these two new full-length sequences from Cabo Verde branched together in a highly supported branch, separate from the other sub-subtype A clusters, and also from the CRF26_AU (Figure 1). Bootscan analysis including all HIV-1 subtypes and A sub-subtypes was conducted and showed that the majority of the studied genomes presented high similarity among them and with no other subtype or sub-subtype. Taking these results together, we could denominate then as a new sub-subtype A: A8 (Figure 2).

These sequences were submitted to BLAST search analysis and retrieved sequences with up to 89.4% of homology with a query cover of 100%. After pol (2253–3218 of the HXB2 reference) BLAST analyses, those 100 sequences with homology higher than 92.4% were included at the pol alignment and the ML tree was performed. However, only five sequences confirmed with a high support value for clustering in a monophyletic clade with the new sub-subtype A, A8. One was sequenced in Portugal (PT), one in the United States (US), one in the Democratic Republic of Congo (DC), one in Spain (ES), and one in Sweden (SE). GenBank accession numbers were as follows: GQ398862, JX460184, MH705159, EF380382 and AY165240, respectively (Figure 3). Those sequences presented BLAST homology above to 94.83% with a query cover of 100% (PT-97.1%, US-97%, DC-95.97%, ES-95.24%, and SE-94.83%) and investigated sequences without origin information. 

### 3.3. Genetic Distance

The intragroup A8 genetic distance varied between 4.0% and 7.4% in the analyzed fragments (complete genome, gag, pol, and env). The distance between the A8 and other sub-subtypes A varied on average from 9.6% (A3) to 12.1% (A2) for gag, from 8.1% (A3) to 10.8% (A2) for pol, from 14.7% (A3) to 19.0% (A4) for env, and from 11.5% (A3) to 14.8% (A2) for the complete genome (Table 2).

## 4. Discussion

The global spread of HIV-1 group M in the second half of the 20th century has led to a complex and constantly changing distribution of subtypes and recombinant forms. Subtype A is responsible for about 10% of the HIV-1 infections worldwide, being found mainly in East Africa [19]. The characterization of the first A sub-subtypes occurred in 2001, distinguishing A1 and A2 [20]. Until 2016, six A sub-subtypes (A1–A4, A6) had been described, and the sub-subtype A5 was found only in the recombinant form CRF26_A5U [7,8,21]. In 2018, a phylogenetic analysis based on public complete genome sequences was carried out and a reclassification of HIV-1 sub-subtypes A was proposed. Thus, subtype A was subdivided into six sub-subtypes (A1–A4, A6, A7), in addition to CRF26_A5U [5,7,8]. Moreover, in 2018, in contrast to the classification adopted for Los Alamos and these studies, a group of researchers stated that on an evolutionary scale, A3 would be part of the sub-subtype A1 clade [22]. In that study, they named A8 the sub-subtype A that took part in CRF36 (composed by CRF01, CRF02, A, G) and CRF37 (composed by CRF01, CRF02, A, G, U). However, as Los Alamos named it A, we designated our sequences as A8 [22].

The HIV-1 sub-subtypes A have dispersed around the world. Inspecting complete genome sequences at the Los Alamos database, it was possible to identify that A1 is the sub-subtype with the largest number (≈80%) of described A sequences and is found mainly in Rwanda and Kenya; A2 and A4 are restricted to the Democratic Republic of Congo; A3 and A7 to Senegal; and the sub-subtype A6 is more often found in the Russian Federation [5].

The high diversity of subtype A, besides being related to its division in sub-subtypes, is also present in the classification of circulating recombinant forms. Among the 100 CRFs already described until January 2021, subtype A is present in 59, some of them being part of a complex CRF involving three or more subtypes or CRF01 or CRF02 and other subtypes [4]. Among the circulating recombinant forms, the most widespread in the world are CRF01_AE and CRF02_AG, which are responsible for 5.3% and 7.7% of HIV-1 group M infections, respectively [19].

Similar to a recent study that shows unique drug resistance profiles of the subtype A6 circulating in former Soviet bloc countries, to reverse transcriptase inhibitors (A62V_RT_ and G190S_RT_) [23], in sub-subtype A8 we verified the presence of L10I/V minor protease inhibitor DRM.

The high diversity of HIV-1 described in Cabo Verde may be related to its proximity and relationships with West Africa and European countries. Studies already conducted show strong similarity to the HIV-1 subtype G between sequences from Portugal and Cabo Verde and propose that historical and recent movements between Angola, Cabo Verde, and Portugal may have played a key role in the origin and dispersion of certain viral clades [24]. Due to its geographical location and investment in tourism, Cabo Verde receives thousands of tourists from all continents, especially Europe. Many foreigners have also come to Cabo Verde as a country of emigration. In 2013, immigrants represented about 3.5% of the total resident population. The majority come from the African continent, 38% from Economic Community of West African States (ECOWAS) countries and 34% from other African countries. This phenomenon of people movement, whether by tourism or emigration, may affect the dispersion and viral diversity of HIV-1 [25].

The continuous monitoring of molecular epidemiology in Cabo Verde and worldwide is extremely important since HIV diversity can impact diagnosis, viral load measures, drug resistance, responses to antiretroviral treatment, pathogenesis, vaccine design, immune response, and viral escape [19].

## 5. Conclusions

The proximity to HIV endemic regions where HIV-1 sub-subtypes A circulate, with a high level of people mobility, whether by tourism or emigration may have contributed to the emergence of sub-subtype variant A8 circulating in Cabo Verde. Further studies with recent samples will be of relevance to assess the dispersion and the role of HIV-1 sub-subtype A8 in the HIV/AIDS epidemic in Cabo Verde.

## Figures and Tables

**Figure 1 viruses-13-01093-f001:**
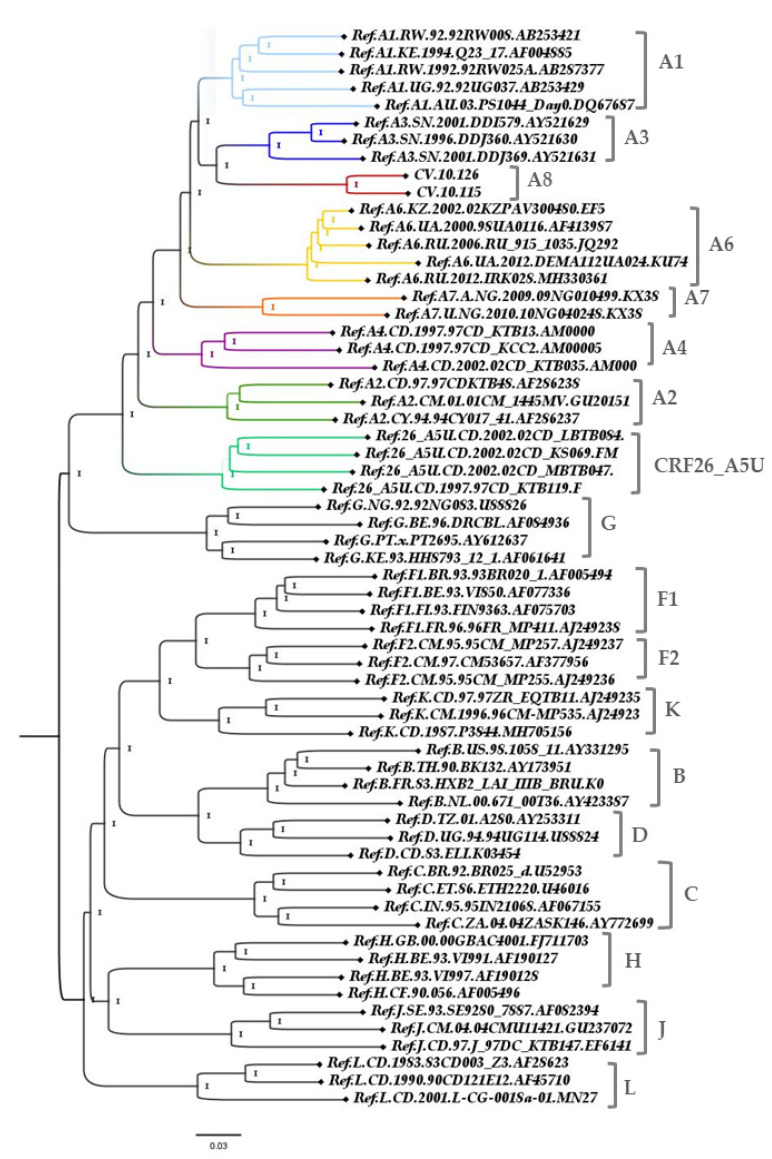
Maximum likelihood (ML) tree implementing nucleotide substitution model General Time Reversible (GTR), indicating the phylogenetic relationships between A sub-subtypes. Approximate likelihood ratio test (ALRT) values are represented only if >0.90. Complete genome analysis (803-9496 relative to the HXB2 reference). The sub-subtype A branches are highlighted.

**Figure 2 viruses-13-01093-f002:**
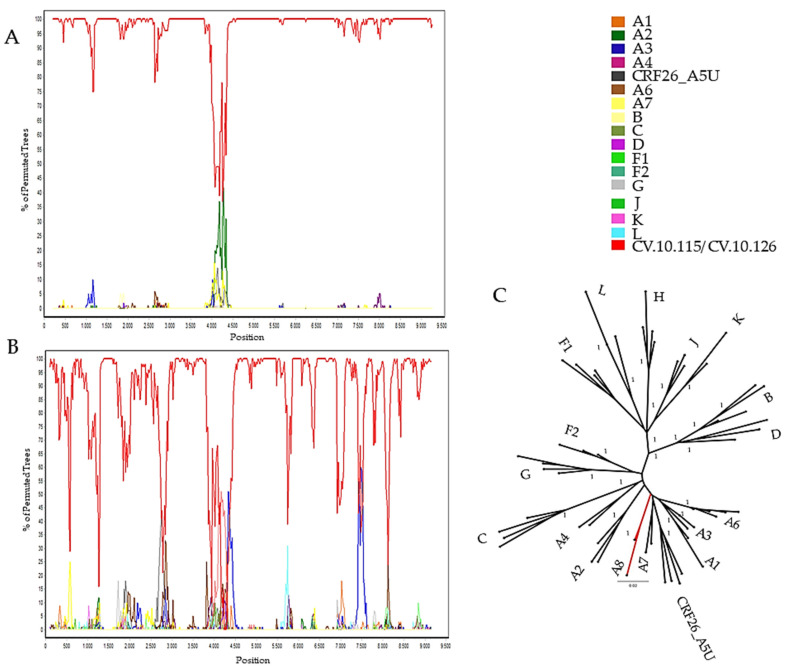
Investigation of genome recombination. Bootscan analysis was performed using a sliding window of 400 nt by increments of 20 nt. (**A**) CV.10.115, (**B**) CV.10.126, and (**C**) Maximum likelihood (ML) tree was implemented for fragments measured when bootscan analysis was above 70% of permuted trees. ML implementing nucleotide substitution model GTR was generated from the alignment position 4000 to 4500, supporting the classification of sub-subtype A8 (in red). ALRT values were represented only if >0.90. Reference sequences GenBank accession number as follows: Ref.B-K03455,AY423387, AY173951, AY331295; Ref.C-U52953,U46016,AF067155,AY772699; Ref.D-K03454, AY253311, U88824; Ref.F1-AF077336, AF005494,AF075703AJ249238; Ref.F2-AJ249236,AJ249237,AF377956; Ref.G-AF084936, AF061641,U88826,AY612637; Ref.H-AF190127,AF190128,AF005496,FJ711703;Ref.J-EF614151,GU237072, AF082394; Ref.L-AF286236,AF457101, MN271384; Ref.K-AJ249239,MH705156,AJ249235; Ref.A1-DQ676872,AB253421, AB253429, AF004885,AB28737; Ref.A2-AF286238, GU201516,AF286237; Ref.A3-AY521630,AY521629,AY521631; Ref.A4-AM000053, AM000054, AM000055; Ref.26_A5U-FM877780, FM877777,FM877782,FM877781;Ref.A6-EF589043,JQ292897, MH330361,AF413987, KU749403;Ref.A7-KX389608,KX389622.

**Figure 3 viruses-13-01093-f003:**
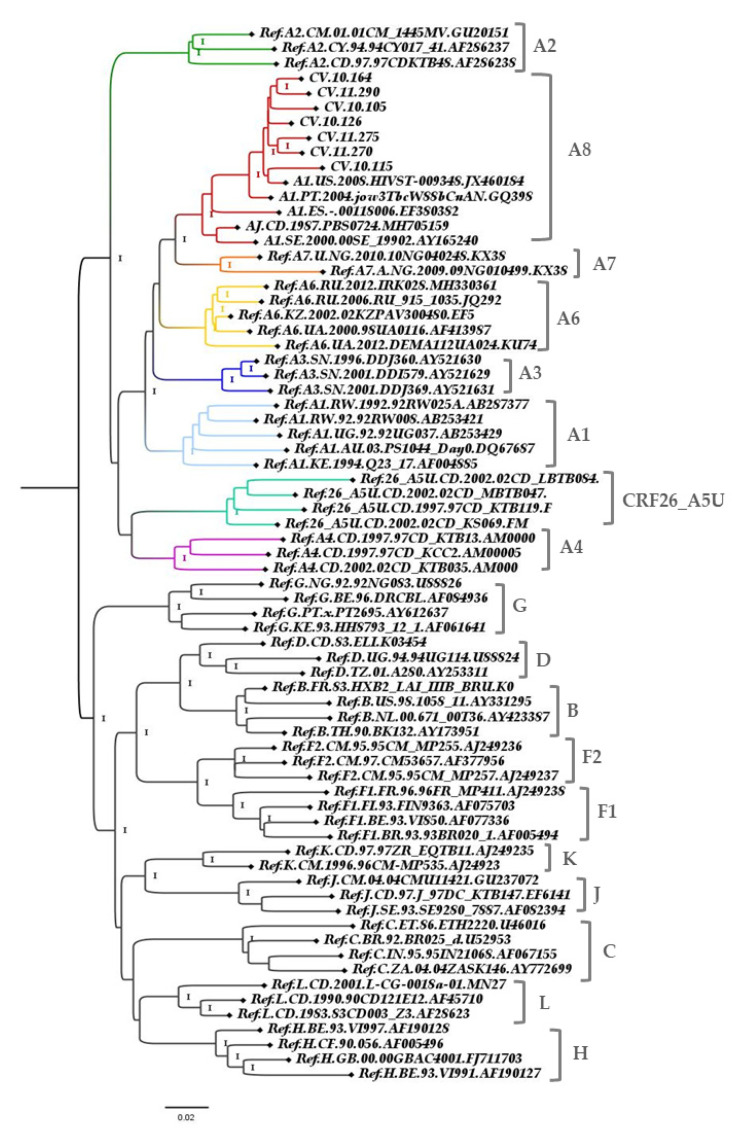
ML tree implementing nucleotide substitution model GTR, indicating the phylogenetic relationships between A sub-subtypes. ALRT values are represented only if > 0.90. Pol fragment (2253–3218 relative to the HXB2 reference). The sub-subtype A branches are highlighted.

**Table 1 viruses-13-01093-t001:** Sociodemographic and clinical data of the studied individuals.

Sample	Island	Year of Diagnosis	Year of Collection	Gender	Age	ARV Status	DRM	GenBank Accession
CV.10.105	Santiago	-	2010	Female	19	treated	NNRTI-E138A	KJ395593 (*pol*)/MW353966 (partial genome)
CV.10.115	Santiago	2001	2010	Female	41	treated	NNRTI-M230I	KJ395594 *(pol*)/MW353967 (complete genome)
CV.10.126	Santiago	2007	2010	Female	44	treated	None	KJ395596 (*pol*)/MW353968 (complete genome)
CV.10.164	Santiago	2009	2010	Male	40	naive	None	KJ395613 (*pol*)
CV.11.270	Santiago	1995	2011	Female	52	treated	None	KJ395697 (*pol*)/MW353969 (partial genome)
CV.11.275	Santiago	-	2011	Female	45	naive	None	KJ395701 (*pol*)
CV.11.290	Sal	2009	2011	Male	44	naive	None	KJ395705 (*pol*)/MW353970 (partial genome)

ARV–antiretroviral; DRM- drug resistance mutation.

**Table 2 viruses-13-01093-t002:** Genetic distance between intragroup A8 and the other HIV-1 A sub-subtypes.

Genetic Distance Average (%)							
	Intragroup	Inter-Sub-Subtype A X A8					
	A8 (*n* = 2)	A1 (*n* = 5)	A2 (*n* = 5)	A3 (*n* = 3)	A4 (*n* = 3)	A6 (*n* = 5)	A7 (*n* = 2)
Complete Genome (803 → 9496)	5.2	12.2 (11.9–12.6)	14.8 (14.1–15.4)	11.5 (11.2–12.0)	14.0 (13.3–14.7)	12.8 (12.2–13.8)	14.0 (13.6–14.5)
GAG (803 → 2280)	4.2	9.9 (8.5–11.9)	12.1 (11.0–13.8)	9.6 (8.5–10.3)	11.9 (11.5–12.8)	10.2 (9.7–13.0)	11.3 (10.5–12.3)
POL (2065 → 5100)	4.0	8.8 (8.5–9.4)	10.8 (10.5–11.1)	8.1 (7.6–8.8)	9.9 (9.2–10.6)	8.6 (7.8–9.7)	8.5 (7.9–9.2)
ENV (6223 → 8804)	7.4	16.0 (14.9–16.8)	18.4 (17.5–19.5)	14.7 (14.1–15.0)	19.0 (17.6–20.4)	17.7 (16.8–17.4)	18.8 (18.4–19.1)

## Data Availability

Nucleotide sequences obtained at the present work are available from the GenBank database (https://www.ncbi.nlm.nih.gov/genbank/ accessed on 17 March 2021) under accessions (MW353966–MW353970).

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
