# Peer review of "Genetic Characterization of a New HIV-1 Sub-Subtype A in Cabo Verde, Denominated A8"

_viruses, 2021, doi:10.3390/v13061093_

Round 1

Reviewer 1 Report

This manuscript characterized a novel subtype A variant circulating in Cabo Verde. The study was interesting showing the enormity of HIV strain diversity in Cabo Verde. The existence of seven individuals with a distinct subtype A subcluster illustrates a distinct circulating founder variant. The findings are noteworthy and the addition to Gen Bank as distinct subtype A variants is important. Recent studies show unique drug resistance profiles of the subtype A6 circulating in former Soviet bloc countries.

The proximity to HIV endemic regions, including West Africa and Congo may have contributed to a novel subtype A variant. However, the study is premature to establish the widespread global existence and characterization of a novel sub-subtype A8 variant.  

Comments

  1. The title should make mention of the geographic location. Cabo Verde.
  2. Although this is a new subtype A variant, classification as A8 requires Gen Bank or Los Alamos HIV database designation.    

Author Response

Thank you very much for the possibility to resubmit our manuscript after revision according to the comments from the reviewers. As you will see, we have addressed and performed below all the suggestions/modifications made by the reviewers to our manuscript, and we hope it will be considered for publication in Viruses. In expectation of a favorable response, we are ready to clarify any additional issues required for the proper evaluation of this manuscript to be considered for publication. English language and style were verified at https://www.mdpi.com/authors/english, previously to the article submission, as requested by the editor.

Point 1: The title should make mention of the geographic location. Cabo Verde. We agree with the referee comment and the title was revised to "Genetic Characterization of a new HIV-1 sub-subtype A in Cabo Verde, denominated A8."

Point 2: Although this is a new subtype A variant, classification as A8 requires Gen Bank or Los Alamos HIV database designation.

Our analyses were submitted to the Los Alamos HIV sequence database, and they agree with us that we had detected a new subsubtype A, and they designated it as A8, as can see below.

"Thanks for sending the sequences.  My preliminary look at them indicates that this is a legitimate new subsubtype. And A8 is be the correct label." This is a part of the returned email by Brian Foley, from Los Alamos HIV sequence database.

Point 3: Recent studies show unique drug resistance profiles of the subtype A6 circulating in former Soviet bloc countries.

As suggested by the referee we included a paragraph in the discussion section concerning the resistance profiles detected in the PR/RT region, as follows: "Similar to a recent study that shows unique drug resistance profiles of the subtype A6 circulating in former Soviet bloc countries, to reverse transcriptase inhibitors (A62VRT and G190SRT) [23], in sub-subtype A8 we verified the presence of L10I/V, minor protease inhibitor DRM."

Point 4:  Are the conclusions supported by the results? (X) can be improved. Thus, we revised this section of the manuscript to focus on the result. The new version of the conclusion section is as follows: " The proximity to HIV endemic regions where HIV-1 sub-subtypes A circulates, high level of people mobility, whether by tourism or emigration may have contributed to the emergence of sub-subtype variant, A8 circulating in Cabo Verde. Further studies with re-cent samples will be of relevance to assess the dispersion and the role of HIV-1 sub-subtype A8 in the HIV/AIDS epidemic in Cabo Verde.”

We also moved the following paragraph “The continuous monitoring of molecular epidemiology in Cabo Verde and worldwide is extremely important since HIV diversity can impact diagnosis, viral load measures, drug resistance, responses to antiretroviral treatment, pathogenesis, vaccine design, immune response, and viral escape [19]” to the discussion section that was improved by the inclusion of the following information “Due to its geographical location and investment in tourism, Cabo Verde receives thou-sands of tourists from all continents, especially Europe. Many foreigners have also come to Cabo Verde as a country of emigration. In 2013, immigrants represented about 3.5% of the total resident population. The majority come from the African continent, 38% from ECOWAS countries and 34% from other African countries (https://ine.cv/wp-content/uploads/2016/10/IMC-2013-Migracoes.pdf). This phenomenon of people movement, whether by tourism or emigration, may affect the dispersion and viral diversity of HIV-1.

Reviewer 2 Report

The present study has identified a new sub-subtype variant of HIV-1, denoted A8, from Cabo Verde in West Africa based upon genetic analysis across the pol and gag sequences and comparisons with other A subtype isolates and the HXB2 reference genome. Overall, this is a nice comprehensive study that highlights the genetic varience and diversity between the different circulating subtypes of HIV-1 on the African continent. The manuscript is well-written and the data clearly presented; and this work will advance our understanding of the molecular evolution of HIV-1.      

Author Response

Thank you very much for the possibility to resubmit our manuscript after revision according to the comments from the reviewers. As you will see, we have addressed and performed all the suggestions/modifications made by reviewer 1 to our manuscript (attached file), and we hope it will be considered for publication in Viruses. In expectation of a favorable response, we are ready to clarify any additional issues required for the proper evaluation of this manuscript to be considered for publication. English language and style were verified at https://www.mdpi.com/authors/english, previously to the article submission, as requested by the editor.
